# A non-canonical lymphoblast in refractory childhood T-cell leukaemia

Bram S. J. Lim [1,15], Holly J. Whitfield [1,15], Mi K. Trinh [1], Gianna Bloye[2], Rebecca Thomas[3], Nathaniel D. Anderson [1], Anna Wenger [1], Angus Hodder[1,3], Taryn D. Treger[1,4,5], Henry Lee-Six [1,4], Tim H. H. Coorens [6], Conor Parks [1], Toochi Ogbonnah[1], Petri Pölönen [7], Charles G. Mullighan [7], David T. Teachey [8], Jason Xu[9], Kai Tan [8,9], Melanie Hagleitner[10], Lennart Kester [10], Frank N. van Leeuwen [10], Gordon Beattie[11,12], Marc R. Mansour [2,13], Owen Williams [13], Jack Bartram [3], Stuart Adams [3], Laura Jardine [1,14], Sam Behjati [1,4,5] ✉ & David O'Connor [2,3,13] ✉

Refractory cancers may arise either through the acquisition of resistance mechanisms or represent distinct disease states. The origin of childhood T-cell acute lymphoblastic leukaemia (T-ALL) that does not respond to initial treatment, i.e. refractory disease, is unknown. Refractory T-ALL carries a poor prognosis and cannot be predicted at diagnosis. Here, we perform single cell mRNA sequencing of T-ALL from 58 children (84 samples) who did, or did not respond to initial treatment. We identify a transcriptionally distinctive blast population, exhibiting features of innate-like lymphocytes, as the major source of refractory disease. Evidence of such blasts at diagnosis heralds refractory disease across independent datasets and is associated with survival in a large, contemporary trial cohort. Our findings portray refractory T-ALL as a distinct disease with the potential for immediate clinical utility.

Acute lymphoblastic Leukaemia (ALL) is the most common childhood malignancy. It is predominantly of B-cell lineage (B-ALL) but has a T-cell phenotype (T-ALL) in 15% of cases[1,2]. T-ALL is a more aggressive and less curable disease, with higher rates of induction failure, early relapse, central nervous system disease and ultimately death than B-ALL[3]. Furthermore, unlike B-ALL, there are no clinical, phenotypic or genetic features that reliably identify children with T-ALL who have a poor prognosis at diagnosis, precluding a risk-adapted approach at the outset of treatment. Despite various proposed diagnostic biomarkers

to separate high-risk from non-high-risk T-ALL, none have been sufficiently robust in replication cohorts to enter clinical practice[4–9].

An elevated level of residual disease after 4 weeks of induction chemotherapy is the earliest reliable indicator of high-risk T-ALL. In particular, children with so-called refractory disease (induction failure), defined as the presence of ≥5% cancer cells (blasts) in the bone marrow following induction treatment[10], constitute a very high-risk disease group. Refractory disease occurs in around 10% of children with T-ALL, of whom only half will survive beyond 5 years[11]. Refractory

[1]Wellcome Sanger Institute, Hinxton, UK. [2]Department of Haematology, UCL Cancer Institute, London, UK. [3]Department of Haematology, Great Ormond Street Hospital for Children, London, UK. [4]Cambridge University Hospitals NHS Foundation Trust, Cambridge, UK. [5]Department of Paediatrics, University of Cambridge, Cambridge, UK. [6]Broad Institute of MIT and Harvard, Cambridge, MA, USA. [7]Department of Pathology and Center of Excellence for Leukemia Studies, St. Jude Children's Research Hospital, Memphis, TN, USA. [8]Children's Hospital of Philadelphia, Philadelphia, PA, USA. [9]Perelman School of Medicine, University of Pennsylvania, Philadelphia, PA, USA. [10]Princess Maxima Center for Pediatric Oncology, Utrecht, The Netherlands. [11]CRUK City of London Centre Single Cell Genomics Facility, UCL Cancer Institute, London, UK. [12]Bioinformatics Hub, UCL Cancer Institute, University College London, London, UK. [13]UCL Great Ormond Street Institute of Child Health, London, UK. [14]Biosciences Institute, Newcastle University, Newcastle upon Tyne, UK. [15]These authors contributed equally: Bram S. J. Lim, Holly J. Whitfield. ✉e-mail: sb31@sanger.ac.uk; david.oconnor@ucl.ac.uk

disease is therefore a major contributor to the overall mortality from childhood T-ALL[12].

Large scale genomic studies have substantially furthered our understanding of the genetic and phenotypic basis of childhood ALL, including T-ALL[13–15]. However, despite extensive investigation, the nature and origin of refractory T-ALL remains largely elusive. Refractory blasts might directly derive from the initial diagnostic blast population as treatment-resistant clones. Alternatively, it is conceivable that refractory blasts represent a distinctive, intrinsically-resistant cancer cell type[16]. Given the scale of past investigations, including a recent combined genetic, transcriptomic and proteomic study of T-ALL with over 1300 participants[17], it would seem improbable that interrogating ever larger cohorts with the same approach will fundamentally shift our understanding of refractory T-ALL. However, unbiased phenotyping of refractory blasts at the resolution of single cells may be more fruitful, especially in the context of recent efforts collectively known as the Human Cell Atlas, that have provided quantitative molecular definitions of normal cell states during T-cell development[18–21]. Accordingly, we set out to investigate, with single cell mRNA sequencing, whether distinctive blast states may account for refractory childhood T-ALL.

## Results

### A census of T-cell lymphoblasts
Initially, we curated a cohort of 21 children with T-ALL, including 13 with refractory disease, selected to represent a typical spectrum of T-ALL genotypes (Fig. 1A, B and Supplementary Data 1, 2). We collected viable cells from diagnostic bone marrow aspirates of all children. Where feasible, for children with induction failure, we collected viable cells from end of induction (day 28) bone marrow aspirates ($n = 8/13$ children). We subjected all samples to flow cytometry using a 16-marker diagnostic panel (Supplementary Data 3) and to concomitant whole transcriptome and T-cell receptor (TCR) single cell sequencing (Chromium 10X). To avoid biases in cell selection, we did not enrich for blasts. Following standard data processing, we obtained expression profiles of 216,507 cells. We clustered cells using Leiden clustering in Scanpy[22] and annotated the resulting clusters, informed by diagnostic flow profiles of blasts and canonical markers of non-cancer cells (Fig. 1C and Supplementary Fig. 1). Through this process, we identified 160,327 blasts and 56,180 normal cells. As has been consistently observed by us and others, normal cells produced lineage specific clusters derived from a mix of patients, whereas cancer cells formed patient specific clusters[23–25].

Most diagnostic (day 0) and end of induction (day 28) blasts co-clustered for each patient, apart from blasts of two children, P030 and P058. Here, refractory day 28 blasts segregated away from their respective day 0 blasts and, remarkably, co-located, indicating some transcriptional similarity between refractory blasts of these two patients. Overlapping genes that were upregulated in refractory day 28 blasts of each patient compared to all other blasts revealed a common profile that defined day 28 refractory blasts, spearheaded by the *ZBTB16* gene (Fig. 1D and Supplementary Data 4). This is noteworthy as *ZBTB16* (Zinc Finger and BTB Domain Containing 16, also referred to as Promyelocytic Leukaemia Zinc Finger, *PLZF*) is a transcription factor that, amongst other functions, regulates lymphocyte differentiation, operating as a switch that promotes the differentiation of unconventional innate T cells (versus conventional T cells)[26,27]. Of note, in child P058 (Fig. 1E), *ZBTB16* expression was confined to a minority blast population at diagnosis (65 cells at day 0 = 0.67%). At day 28, this diagnostic minority population became dominant, accounting for 97.6% of blasts. Similarly, we identified *ZBTB16* expression as the key difference when comparing all refractory day 28 blasts against all diagnostic blasts in children with responsive disease (Supplementary Data 5, 6).

### *ZBTB16* expression in refractory blasts
Motivated by these findings, we interrogated *ZBTB16* expression across all samples of the discovery cohort (Fig. 1F). In diagnostic (day 0) samples of children with responsive disease, *ZBTB16* expression was largely absent from blasts. By contrast, *ZBTB16*+ blasts and clusters of blasts were common in children with refractory disease, both at diagnosis and at day 28 after induction treatment (i.e. at the point of refractory disease) (Fig. 1G). Overall, our findings indicate that at diagnosis, *ZBTB16* expression may delineate a blast population that resists induction treatment. Importantly, we were able to measure the presence of ZBTB16+ populations by intracellular flow cytometry (Fig. 1H), which validates the mRNA signal and offers a tractable method for measuring ZBTB16+ cells, even at low frequency, in clinical diagnostic practice at the resolution of single cells.

### Refractory blasts exhibit a non-canonical T-cell state
To examine what cell type *ZBTB16*+ blasts represent, we compared blasts with detailed single cell transcriptional definitions of normal immune cells. Broadly, lymphocyte progenitors that generate T cells give rise to conventional T cells of the adaptive immune system and to unconventional innate T cells, as well as to innate-like lymphocytes, comprising natural killer (NK) cells and innate lymphoid cells (ILC) (Fig. 2A). To investigate which normal immune cells resemble *ZBTB16*+ blasts, we directly compared normal single cell transcriptomes[21] against blast transcriptomes, using logistic regression[28] (Fig. 2B). In a comparison of normal cell types against blasts that did or did not express *ZBTB16*, we found that NK cells, ILCs and unconventional innate T cells most closely resembled *ZBTB16*+ blasts. Conversely, conventional T cell subtypes were more similar to canonical *ZBTB16*− blasts. Consistent with these observations, we found a striking asymmetry across blast populations in the expression of canonical, cell type-defining markers of different lymphocyte variants (Fig. 2C), but did not identify any particular TCR gene expression status that correlated with *ZBTB16* expression (Fig. 2D and Supplementary Data 7). We observed the highest and most consistent expression of conventional T cell markers in *ZBTB16*− blasts of children who responded to induction treatment. By contrast, the day 28 *ZBTB16*+ blasts of children with refractory disease showed the highest expression of NK cell, ILC and unconventional innate T cell markers.

A previous report[29] has suggested an association between *ZBTB16* expression and resistance to glucocorticoid treatment in cell line models of ALL (including T-ALL). We therefore assessed signals of non-canonical blasts through a *ZBTB16*-based gene module (Supplementary Fig. 2) in a recently published dataset of drug sensitivity in primary blast cultures of childhood T-ALL[30]. Although in some experiments the number of data points precluded meaningful statistical analyses, we were able to see evidence of a differential response of T-ALL with non-canonical blasts, such as increased resistance to drugs used in induction therapy, including corticosteroids and daunorubicin, in keeping with its association with induction failure (Supplementary Fig. 3).

### Validation in extended institutional cohort
The key implication of our findings is that a distinctive T lymphoblast population may exist that underpins refractory disease. To validate and extend this finding, we assembled a validation cohort of 37 unselected children with T-ALL from our institutional archives and subjected samples from all available disease assessment points (55 samples) to single cell mRNA sequencing (Fig. 3A, Supplementary Fig. 4 and Supplementary Data 1). Whether measuring the presence of non-canonical blasts by *ZBTB16* expression (Fig. 3B) or through unbiased cell to cell matching by logistic regression[28] (Supplementary Fig. 5), our extension cohort confirms our initial finding that non-

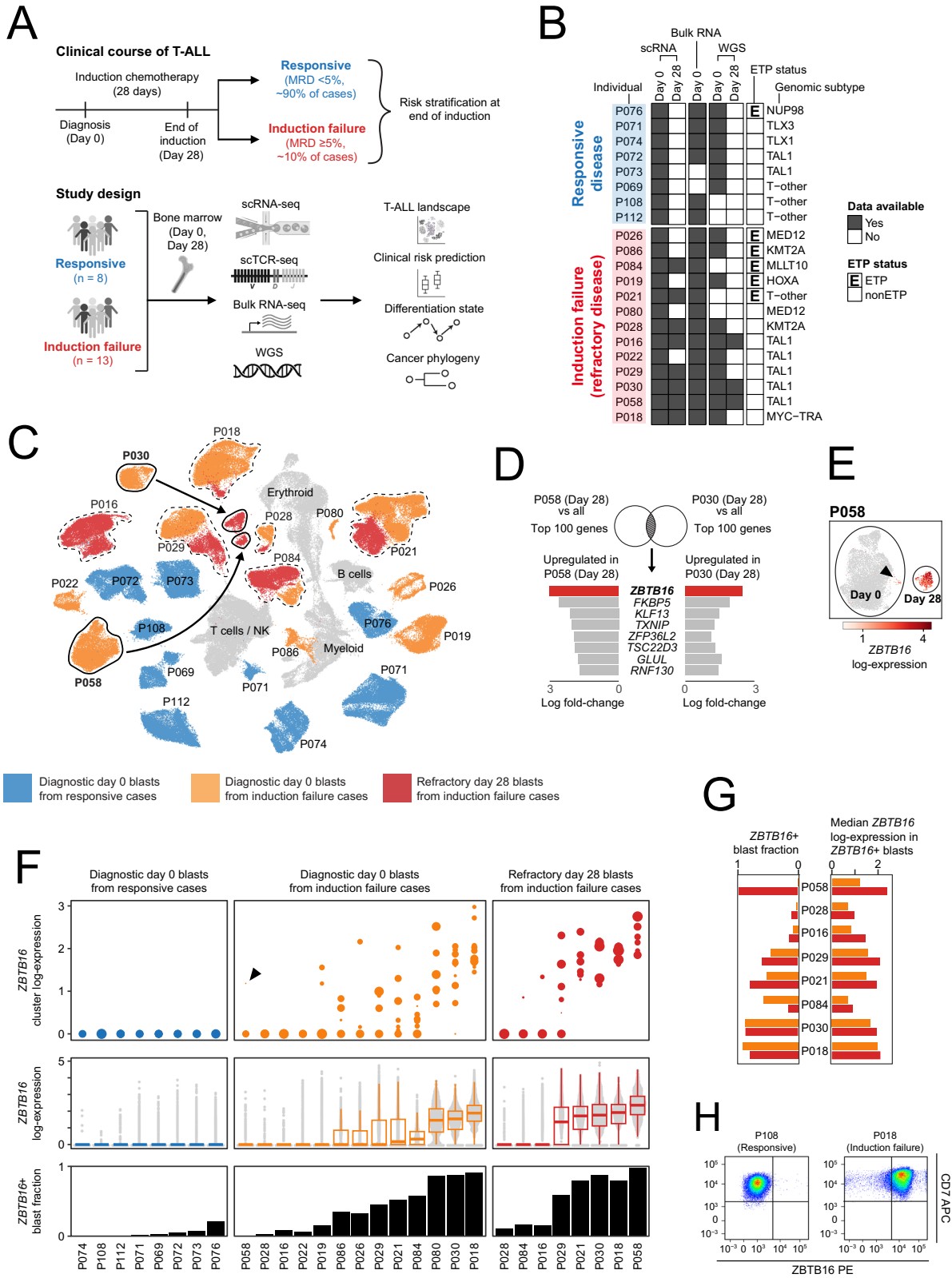

canonical blasts are greatly enriched in refractory disease both at diagnosis ($p = 2.6 \times 10^{-7}$) and at subsequent time points including recurrent disease ($p = 4.4 \times 10^{-9}$). Of note, we also observed an enrichment of non-canonical blasts in cases of responsive but high (1–5%) minimal residual disease at the end of induction ($p = 0.01$). These findings corroborate our initial observation that the presence of *ZBTB16+* blasts at diagnosis may herald induction failure.

## Validation in external clinical cohorts

To further investigate the clinical significance of non-canonical *ZBTB16+* blasts, we extended our enquiry into bulk RNA sequencing data of T-ALL, which enables the assessment of much larger cohorts, albeit with reduced precision. To this end, we derived a 29-gene module for *ZBTB16+* blasts (Supplementary Fig. 2 and Supplementary Data 6, 8), using a pseudo-bulk differential gene expression approach.

**Fig. 1 | *ZBTB16* defines refractory blasts in T-ALL. A** Schematic illustrating the clinical course of T-ALL and overview of study design. Created in BioRender. Behjati, S. (2025) https://www.biorender.com/4eu223k. **B** Heatmap indicating data generated for each patient and clinical information (ETP status, induction outcome, genomic subtype). **C** UMAP (uniform manifold approximation and projection) of 216,507 cells, including 160,327 leukaemia blasts (coloured) and 56,180 normal cells (grey). Day 0 blasts from patients who responded to induction treatment (blue); day 0 blasts from patients with induction failure (orange); day 28 refractory blasts from patients with induction failure (red). Circles and arrows highlight that day 28 blasts from patients P030 and P058 clustered separately from their respective day 0 blasts on the UMAP. **D** Analytical schematic. Overlapping the top 100 upregulated genes of the day 28 blasts from the two patients (P030 and P058) yielded eight genes, of which *ZBTB16* had the highest log-fold increase. **E** UMAP showing the expression of *ZBTB16* in day 0 and day 28 blasts from P058. *ZBTB16* expression is largely absent in day 0 blasts, apart from a small cluster (65/9668 blasts) indicated by an arrowhead, whereas most day 28 blasts expressed *ZBTB16* (692/709 blasts). **F** *ZBTB16* expression in blasts across 29 samples from 21 children with T-ALL. Samples comprise diagnostic day 0 blasts from responsive cases (8 children), diagnostic day 0 blasts from induction failure cases (13 children) and refractory day 28 blasts from induction failure cases (8 children). Top: Median *ZBTB16* expression of each cluster of blasts within each sample; size of circle indicates cluster size; arrowhead indicates a small cluster of day 0 blasts from P058 expressing *ZBTB16*. Middle: Box plots and swarm plots of *ZBTB16* expression at single-cell resolution. Box plots show the first and third quartiles (boxes), as well as median values (central lines). Whiskers extend to the most extreme values within 1.5 times the interquartile range above and below the boxes. Bottom: Fraction of blasts expressing *ZBTB16*. **G** *ZBTB16* expression in blasts from induction failure patients with paired day 0 (orange) and day 28 samples (red). Left: Fraction of blasts expressing *ZBTB16*. Right: Median *ZBTB16* expression in *ZBTB16*+ blasts. **H** Flow cytometry measurement of ZBTB16 in blasts from diagnostic samples of responsive patient P108 and induction failure patient P018. MRD minimal residual disease, scRNA-seq single cell mRNA sequencing, scTCR-seq single cell T-cell receptor sequencing, WGS whole genome sequencing, ETP early T-cell precursor, NK natural killer cell. Source data are provided as a Source Data file.

The module contains genes that are upregulated in *ZBTB16*+ blasts compared to *ZBTB16*− blasts and represents a broader cell state that may not be captured by *ZBTB16* alone. We tested both this module and *ZBTB16* expression alone in bulk mRNA sequencing data generated from our initial samples, demonstrating the validity of both measures for segregating children with and without non-canonical blasts at diagnosis (Supplementary Fig. 2).

Next, we examined different data sets of bulk transcriptomes from children with T-ALL, measuring module score or *ZBTB16* expression on its own. First, we studied 52 bulk transcriptomes obtained from diagnostic samples of children treated at a Dutch paediatric oncology unit (Supplementary Data 9). Grouping cases by response to induction treatment singled out children with refractory disease (i.e. children with ≥5% blasts after induction treatment). There was a significant difference between refractory versus responsive disease (Fig. 4A), whether measured by *ZBTB16* transcript levels ($p < 0.001$) or by module score ($p < 0.001$). Next, we examined data generated from participants of the Children's Oncology Group (COG) AALL0434 trial[17], which represents the world's largest genomic study of T-ALL. We were able to analyse *ZBTB16* expression and module score in 1175 children for whom both metadata and bulk RNA sequencing data (with >60% blast content) from a diagnostic sample were available (Supplementary Data 10). Consistent with our observations thus far, we found significantly raised *ZBTB16* transcript levels ($p < 10^{-12}$) and module scores ($p < 10^{-12}$) in children with refractory disease (Fig. 4B). We then examined the effects of *ZBTB16* expression and module score on survival. Given that we have shown non-canonical blasts are enriched in refractory disease, which is less curable than responsive disease, we would expect to see an association of non-canonical blasts with decreased survival. Accordingly, in survival analyses we found that *ZBTB16* expression ($p < 0.001$) and module score ($p < 0.001$) were associated with both reduced overall and event free survival (Fig. 4C).

### Relation of non-canonical blasts and early T-cell precursor status

Diagnostic immunophenotyping has conventionally categorised T-ALL lymphoblasts into stages parallel to normal thymocyte development using their expression of key surface and cytoplasmic proteins. The only category recognised as a distinct entity since the 2017 WHO classification is early T-cell precursor (ETP) T-ALL, which is defined by flow cytometry via the absence of CD8, CD1a, and CD5, and presence of specific myeloid or stem cell antigens in combination with typical T-ALL antigens (CD7 and cytoplasmic CD3)[31]. Whilst ETP status initially emerged as a marker of high-risk disease[4], subsequent studies using contemporary chemotherapy protocols found it did not predict outcome[32,33]. Importantly, an analysis of ETP status in the aforementioned COG study found no correlation between ETP status and survival[33]. We therefore directly compared the predictive value of the presence of non-canonical blasts at diagnosis against ETP status. A multivariate analysis by Cox proportional hazards showed that, unlike ETP status, evidence of non-canonical blasts conferred a statistically significant effect on survival outcomes (Fig. 4D, Supplementary Fig. 6A and Supplementary Data 11). This observation held true whether we examined the top and bottom tertiles of *ZBTB16*, or treated *ZBTB16* as a continuous variable. Similarly, evidence of non-canonical blasts outperformed a newly devised transcriptional definition of ETP (termed "ETP-like")[17] in predicting overall survival (Supplementary Data 12). Examining the relationship between non-canonical blasts and ETP status in more detail showed that the former's prognostic superiority over ETP derived from increased sensitivity and specificity (Supplementary Fig. 6B). These findings would indicate that non-canonical blasts represent a refractory cell type that is not fully captured by the ETP phenotype.

### Relation of non-canonical blasts and bone marrow progenitor signatures

To identify transcriptomic features of refractory blasts, a recent study derived a set of bone marrow progenitor ("BMP-like") blast signatures[34] that were predictive of poor outcome. Multivariate analysis by Cox proportional hazards showed that the *ZBTB16* module score has a higher predictive ability than the BMP-like signatures in predicting overall survival in the COG cohort (Fig. 4E and Supplementary Data 13). Additionally, using scRNA-seq data from this study[34], we found that *ZBTB16* expression is elevated in patients who had higher minimal residual disease (MRD) at the end of induction (Supplementary Fig. 6C). These analyses show that evidence of non-canonical blasts at diagnosis has prognostic significance, suggesting that they represent a cell type of refractory disease that has not been previously articulated.

### Origin of non-canonical blasts by TCR analysis

Phenotypic cell states may or may not represent the origin of cancers given the inherent plasticity of transcription. Somatic genetic changes, on the other hand, are generally fixed and thus impart a permanent imprint of development. One source of somatic genetic changes in T cells is the rearrangement underpinning T-cell receptors (TCR), which forms the basis for MRD measurements in clinical practice[35]. The gene groups (*TRD*, *TRG*, *TRB*, *TRA*) that form TCRs are rearranged (by deleting genes that are not used) and expressed sequentially during normal T-cell development[36] (Fig. 2A). Their rearrangement and expression, therefore, may represent the point at which the development of emerging blasts deviated from normal differentiation. With

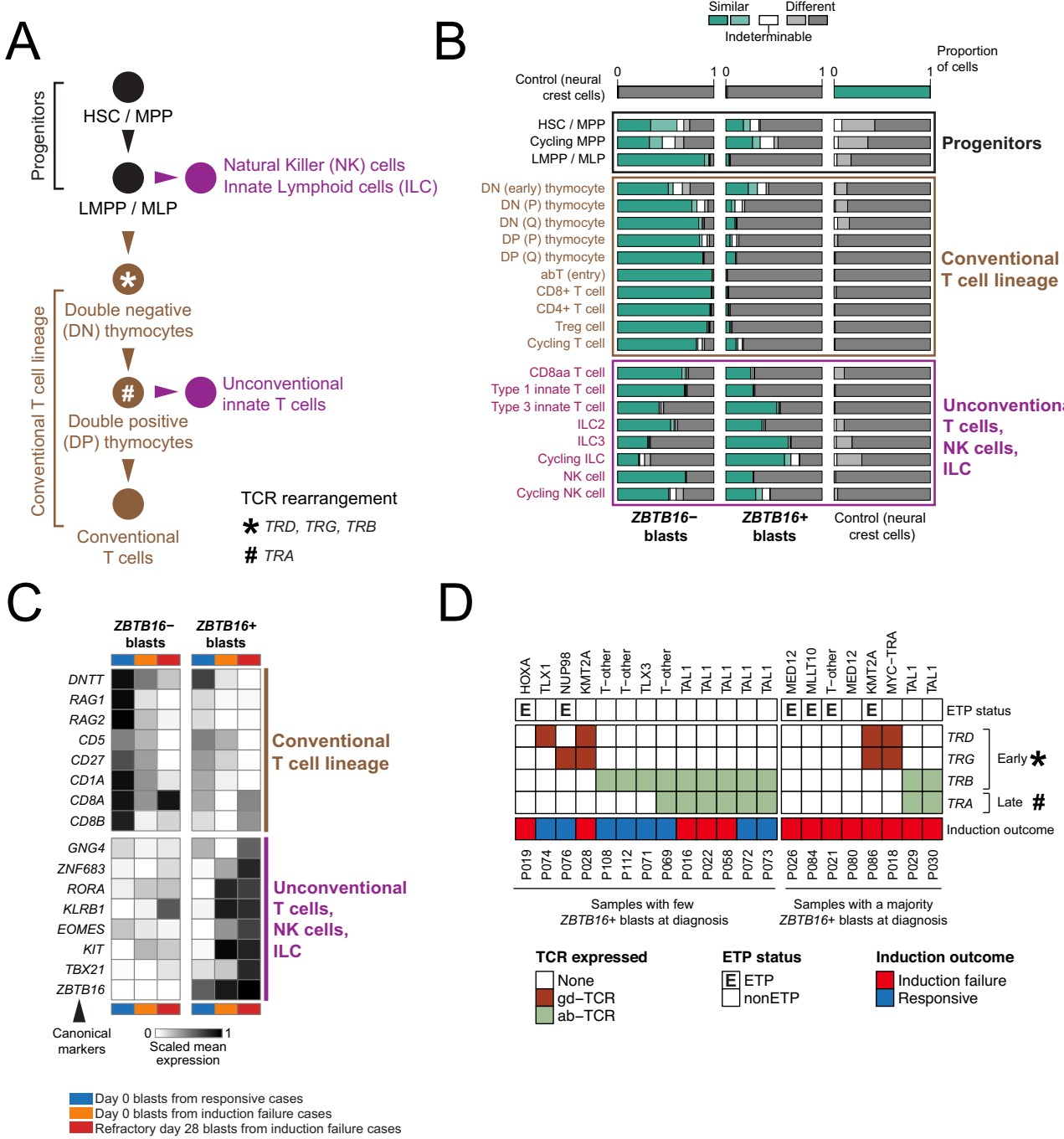

**Fig. 2 | ZBTB16 defines a non-canonical T-ALL cell state. A** Schematic illustrating the differentiation trajectory of T cells, natural killer (NK) cells and innate lymphoid cells (ILC), showing the approximate timepoints for rearrangement of TCR genes (*TRD, TRG, TRB, TRA*). **B** Normal-to-leukaemia transcriptome comparison by logistic regression. A logistic regression model was trained using *ZBTB16*+ blasts, *ZBTB16*− blasts, and neural crest cells which serve as control. This model was used to determine the similarity of query cell types (*y*-axis: conventional T cells, unconventional T cells, NK cells, ILC)[21] to reference cell types (*x*-axis). **C** Heatmap showing expression of well-characterised cell type marker genes across *ZBTB16*+ and *ZBTB16*− blasts, taken from day 0 samples of responsive patients (blue), from day 0 samples of induction failure patients (orange), and from day 28 samples of induction failure patients (red). Expression values are log-normalised gene expression averaged across blasts within each group. Genes are known markers for conventional T cell lineages (top), and unconventional T cells, NK cells and ILC (bottom). **D** Heatmap showing the lack of association between *ZBTB16* expression at diagnosis and TCR gene expression status determined by TRUST4 analysis of diagnostic single-cell mRNA sequencing data. ETP status and genomic subtype are indicated alongside. HSC haematopoietic stem cell, MPP multipotent progenitor, LMPP lymphoid-primed multipotent progenitor, MLP multi-lymphoid progenitor, ETP early T-cell precursor, TCR T-cell receptor. Source data are provided as a Source Data file.

this in mind, we analysed TCR expression across blast populations (Fig. 2D and Supplementary Data 7). In 5 out of 21 children, blasts did not express TCR genes, consistent with a block in the earliest stages of T-cell development, broadly reflected in the ETP status of these

children. Of the eight children with refractory disease and *ZBTB16*+ blasts, TCR expression was otherwise varied, indicating that *ZBTB16*+ blasts may arise from maturation arrest at various stages of T-cell development.

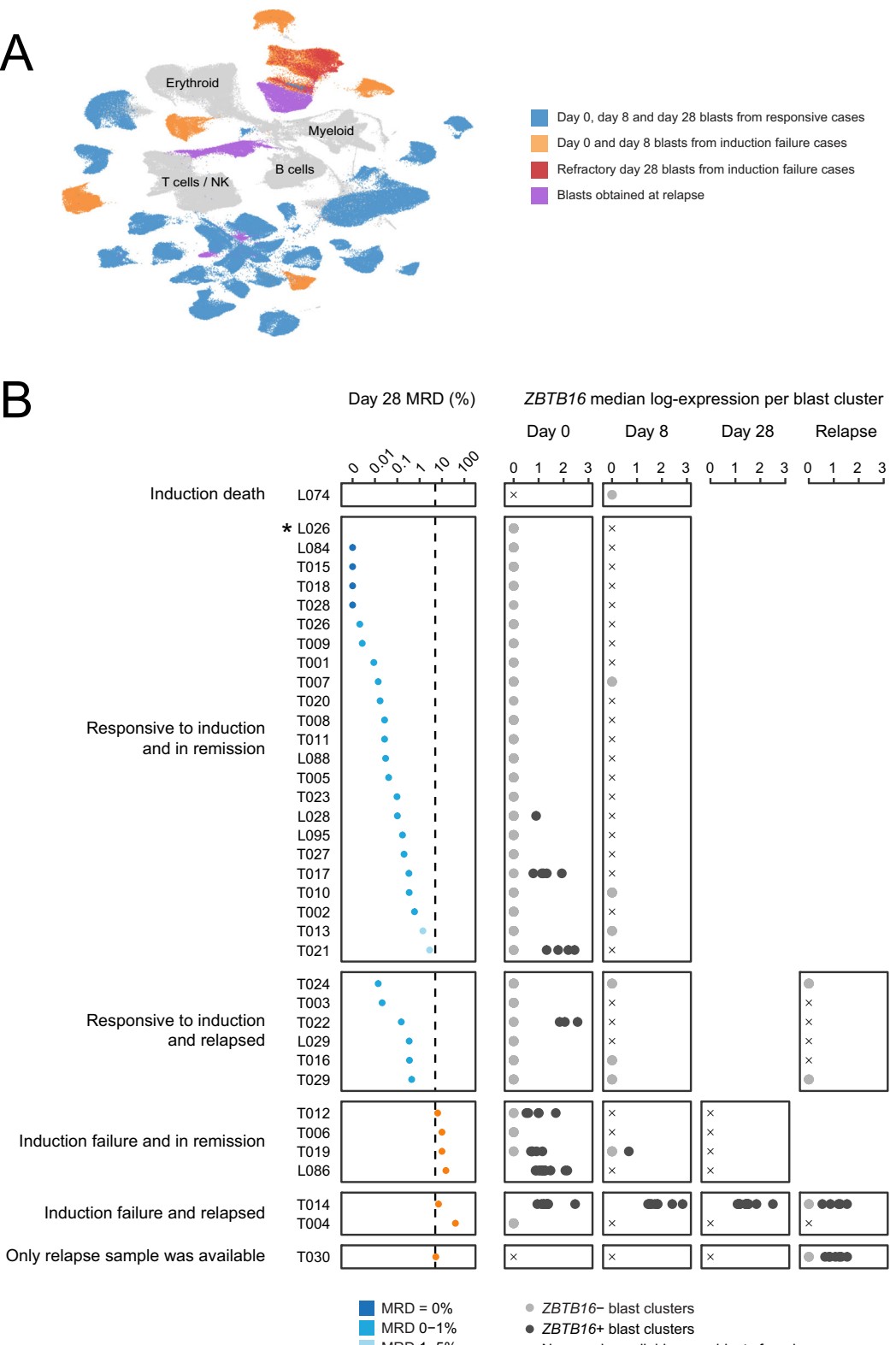

**Origin of non-canonical blasts by phylogenetic reconstruction**
Next, we examined in detail P058 and P030, the index children in whom refractory blasts segregated from diagnostic cells. P058 represented a unique opportunity to determine the phylogenetic relation of sensitive blasts (i.e. *ZBTB16*− blasts that disappeared during induction) and refractory *ZBTB16*+ blasts, enabling us to determine whether *ZBTB16*+ blasts evolved directly from *ZBTB16*− blasts, or represented

an independent, parallel cancer lineage in this child (Fig. 5A). We performed whole genome sequencing (WGS) of diagnostic, refractory and remission samples (to subtract germline mutations). We sequenced the refractory day 28 sample to high depth (429x) in view of the low blast burden (5%). We called all classes of mutations and determined the phylogenetic relation of samples using an established framework[37,38]. In addition, we analysed TCR gene usage from single

**Fig. 3 | *ZBTB16* expression in scRNA-seq of validation cohort. A** UMAP (uniform manifold approximation and projection) of 333,487 cells, including 218,599 leukaemia blasts (coloured) and 114,888 normal cells (grey). Day 0, day 8 and day 28 blasts from patients who responded to induction treatment (blue), day 0 and day 8 blasts from patients with induction failure (orange), day 28 refractory blasts from patients with induction failure (red), and blasts found at relapse (purple). **B** Clinical outcome and *ZBTB16* expression across our extension cohort of 37 unselected children (including longitudinal timepoints where available) from our institutional archives. The first panel from left shows day 28 MRD (%), coloured by MRD group, where patients are split along the *y*-axis by clinical outcome. The following panels show median *ZBTB16* expression of each cluster of blasts within each sample across timepoints; filled circles are coloured grey if median *ZBTB16* expression in that cluster of blasts is zero. L026 is a child with T-lymphoblastic lymphoma who was found by CT scan to have responded completely at day 28. MRD minimal residual disease. Source data are provided as a Source Data file.

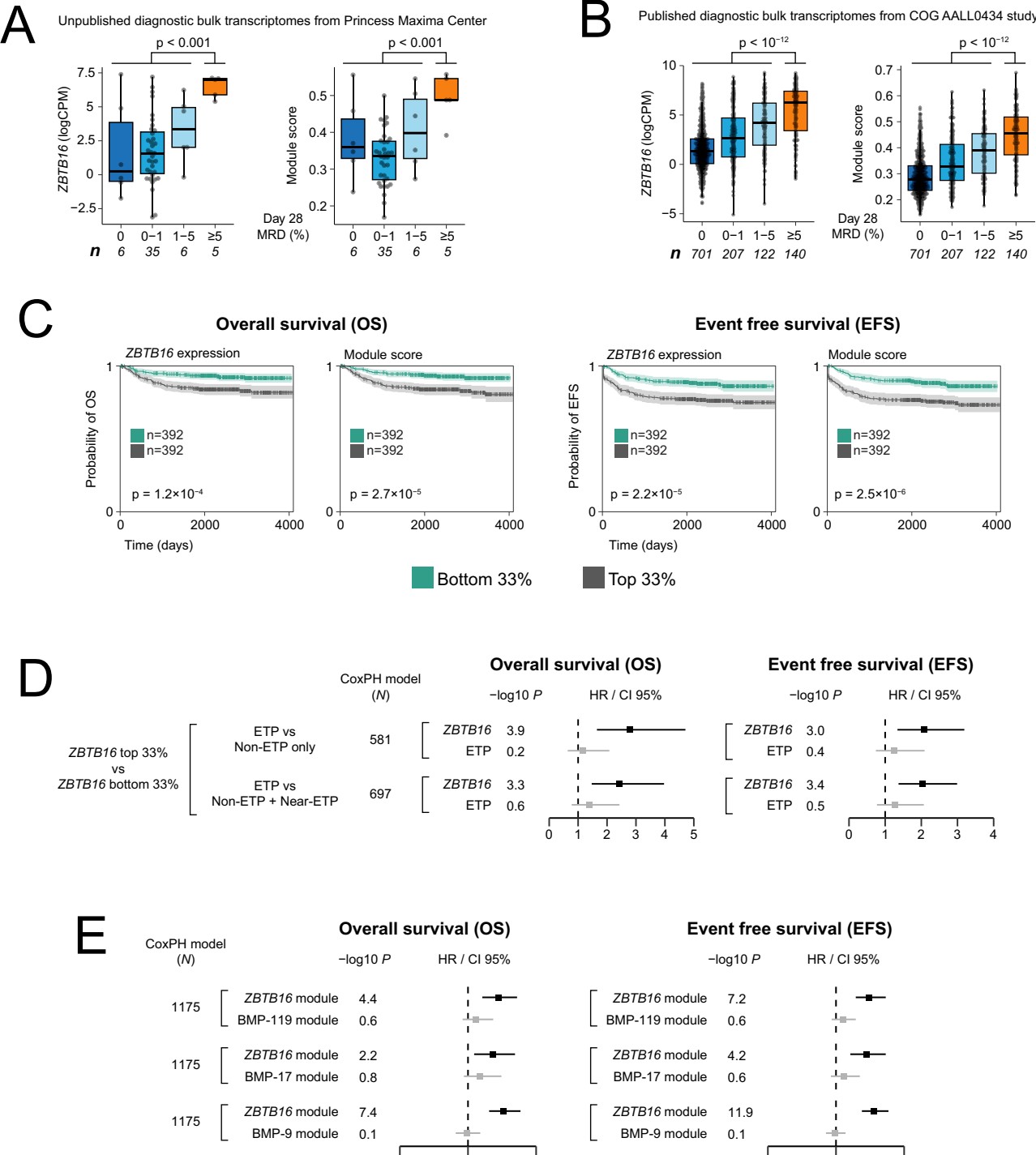

**Fig. 4 | Clinical validation of *ZBTB16* signal in independent T-ALL cohorts. A** Box plots showing *ZBTB16* expression ($p = 2.4 \times 10^{-4}$) and module score ($p = 4.3 \times 10^{-4}$) in bulk transcriptomes of an unpublished Princess Maxima Center cohort, plotted against day 28 MRD categories. The number of individuals per category is indicated by '*n*'. **B** Box plots showing *ZBTB16* expression ($p < 2.2 \times 10^{-16}$) and module score ($p < 2.2 \times 10^{-16}$) in bulk transcriptomes of the published COG AALL0434 study[17], plotted against day 28 MRD categories. The number of individuals per category is indicated by '*n*'. **C** Overall survival and event-free survival in the COG AALL0434 cohort. Data were stratified into the top and bottom 33% of samples based on either their *ZBTB16* expression or module score to test the association of these groups with overall survival and event-free survival using the Kaplan–Meier method. Error bands show 95% confidence intervals (CI 95%). **D** Hazard ratios (HR, central dots) and 95% confidence intervals (CI 95%, error bars) for various Cox proportional hazard (CoxPH) models of overall survival and event-free survival using the published COG AALL0434 study. Both *ZBTB16* and immunophenotype-defined ETP

status were tested as variables, where *ZBTB16* was considered as tertiles as defined in (**C**), and ETP status was considered either excluding or including the "Near-ETP" label. Number of individuals in each CoxPH model is indicated by '*N*'. Statistically significant hazard ratios ($p < 0.05$) in black and non-significant ones in grey. **E** Hazard ratios (HR, central dots) and 95% confidence intervals (CI 95%, error bars) for various Cox proportional hazard (CoxPH) models of overall survival and event free survival using the published COG AALL0434 study. The *ZBTB16* module score was tested against scores for each of the "BMP-like" gene signatures[34]. Number of individuals in each CoxPH model is indicated by '*N*'. Statistically significant hazard ratios ($p < 0.05$) in black and non-significant ones in grey. Box plots show the first and third quartiles (boxes), as well as median values (central lines). Whiskers extend to the most extreme values within 1.5 times the interquartile range above and below the boxes. All box plot *p* values were calculated by a one-sided Wilcoxon rank-sum test. MRD minimal residual disease. Source data are provided as a Source Data file.

---

cell TCR sequences and from single cell mRNA data as independent phylogenetic markers.

Our findings showed that refractory blasts did not derive from diagnostic (treatment sensitive) *ZBTB16*− blasts, but instead evolved from a parallel lineage, as supported by three lines of evidence (Fig. 5B–H and Supplementary Fig. 7). First, TCR gene usage showed that *TRG* genes were common to both blast populations whilst independent *TRB* genes were used by *ZBTB16*− and *ZBTB16*+ blasts (Fig. 5F). Second, analyses of copy number changes, determined from WGS, demonstrated the same phylogenetic relationship. Both samples harboured chromosome 9p copy-neutral loss of heterozygosity (LOH), whilst gain of chromosome 17q was specific to day 0. Despite the low tumour content at day 28, we could demonstrate the absence of the chromosome 17q gain by performing haploblock resolved genotyping to detect allelic imbalances (Fig. 5C). Projecting copy number changes onto single cell mRNA data, using a method that assesses expression-independent allele frequencies[39], confirmed the absence of 17q gain only in *ZBTB16*+ blasts, including the minority population at day 0, whilst all blasts exhibited 9p LOH (Fig. 5D, G). Finally, base substitutions further evidenced the parallel evolution of *ZBTB16*− and *ZBTB16*+ blasts (Fig. 5H and Supplementary Data 14). We did not identify additional genetic driver mutations at day 28 within the constraints of a blast sequencing coverage of 21x (5% of 429x coverage).

P030 was the other child in the discovery cohort whose post-induction blasts clustered separately from diagnostic blasts (Fig. 1C). Here, phylogenetic analysis showed multiple clones at diagnosis and at day 28, which all expressed *ZBTB16* (Supplementary Fig. 8). Consistent with this finding, P030 blasts at both timepoints exhibited the same TCR gene usage (Supplementary Data 7). Therefore, unlike in P058, disease persistence in P030 was not driven by the outgrowth of one particular clone.

**Implications for therapeutic targeting of lymphoblast antigens**

Our discovery of non-canonical lymphoblasts in refractory T-ALL raises immediate questions for current efforts that aim to develop therapies against lymphoblast antigens, in particular, whether the antigen profiles of conventional and non-canonical blasts are similar. We therefore investigated the expression pattern of genes encoding cell surface proteins (Fig. 6). First, we examined the expression of current targets that are under active investigation or in early phase studies[40–44], which showed a mixed picture. CD7, for example, is universally expressed by all blasts and, by definition, by all types of normal T cells, whilst other targets seem relatively sparse in normal conventional T cells. Of note, the expression of two current targets, CD2 and CD5, is decreased in non-canonical blasts. Next, we identified potential targets, leveraging the high resolution of our data in conjunction with reference atlases of normal human T cells[45]. In brief, we searched for genes that were more highly expressed in lymphoblasts compared to normal T cells (Fig. 6A). We found a range of potential targets which exhibited an equal or

more favourable expression pattern in blasts, compared to those of T-ALL targets currently under investigation (Fig. 6B and Supplementary Data 15).

## Discussion

Our investigation of single lymphoblast transcriptomes revealed a non-canonical T-cell blast in refractory T-ALL. We were able to discern the non-canonical cell state with separate methodologies finding clinically relevant effects in independent cohorts with surprising clarity, indicating that non-canonical blasts represent a fundamentally distinctive cell type rather than a subtly altered cell state. Detection of this non-canonical cell type is amenable to measurement by flow cytometry for the ZBTB16 protein. Given the ease of incorporating measurements of cell markers into diagnostic practice through flow cytometry, our findings therefore have the potential to transform risk stratification in T-ALL.

Extensive investigations of T-ALL have failed to identify genetic or phenotypic features that reliably predict response to induction treatment and overall survival. Several diagnostic markers have been described in the past to stratify the treatment of T-ALL[4–9], but no marker has proven sufficiently robust to enter clinical practice. Accordingly, at present, refractory disease cannot be reliably predicted at diagnosis; it can only be identified after the event has occurred. We found that evidence of non-canonical blasts was superior in predicting refractory disease at diagnosis compared to ETP status, ETP-like status or the more recently defined transcriptional BMP signatures. This is even more remarkable given the inherent imprecision of bulk mRNA data that precludes the assessment of individual cells.

Central to our discovery was the expression of *ZBTB16* by non-canonical blasts. The exquisite sensitivity and specificity of a single gene for a cell type is unexpected and is probably explained by the physiological role of *ZBTB16*. In normal T-cell development, expression of this gene operates as a switch that shifts T-cell differentiation away from conventional T cells and towards innate-like lymphocytes. Accordingly, experimental over-expression of *ZBTB16* is sufficient to induce innate-like activation and cytokine expression features even in mature conventional T cells[46]. *ZBTB16* expression has emerged from independent experiments in leukaemia cell lines as a marker of drug resistance[29]. *ZBTB16* itself, and its downstream effectors, may therefore represent targets for eradicating *ZBTB16*+ blasts. Although *ZBTB16* may act as a lineage switch, it is possible that it ceases to be functional when non-canonical blasts present as disease. Whether *ZBTB16* induces the non-canonical blast or is simply a non-functional marker of it, the *ZBTB16*+ non-canonical blast represents a prognostic cell type that is distinct from existing markers of refractory disease.

Our phylogenetic analyses indicate that non-canonical blasts do not arise from a fixed point of T-cell differentiation, which would argue against a strict hierarchy in the development of non-canonical blasts. In one child, in whom non-canonical blasts evolved during induction

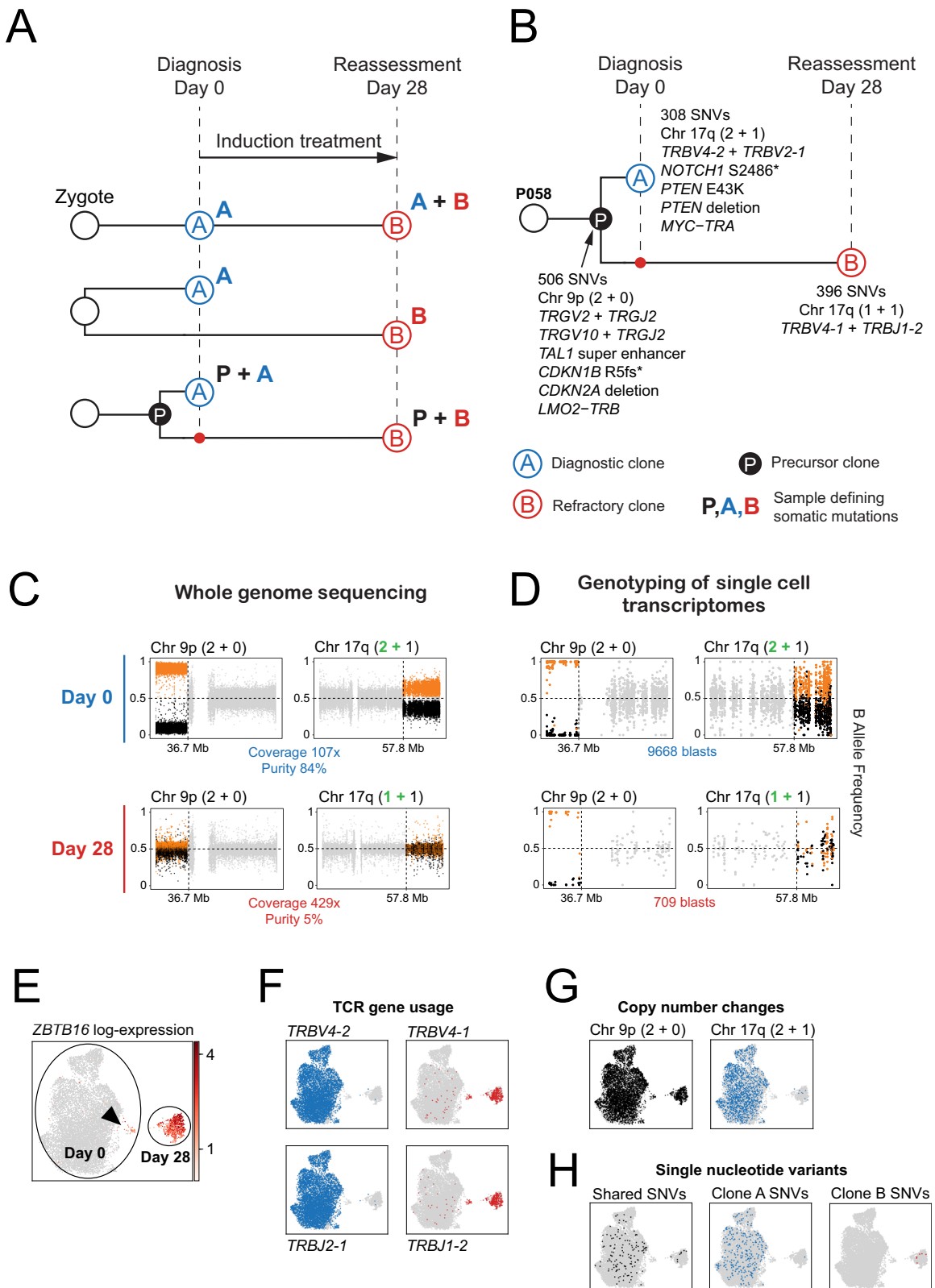

and were present only in miniscule numbers at diagnosis, we were able to provide robust evidence, by projecting genetic variants onto single cell transcriptomes, that non-canonical blasts evolved in parallel to conventional blasts. What causes non-canonical blasts to assume an innate-like lymphocyte identity remains unclear. Given that a decade of extensive omic analyses of T-ALL have failed to associate any specific genetic variant with refractory disease, it is possible that

characteristics that elude current assays, or chance, determine the differentiation fate of evolving lymphoblasts.

It may be worth reflecting on why our work enabled the discovery of non-canonical blasts as the likely basis of many cases of refractory T-ALL. The key to our discovery, beyond its single cell resolution, was inclusion of day 28 samples, i.e. the actual refractory blasts themselves. In addition, comparing our design to other single cell mRNA work, we

**Fig. 5 | Origin of *ZBTB16*+ blasts in patient P058 with refractory T-ALL.**
**A** Possible cancer phylogenies relating clone A at diagnosis (day 0) and clone B at reassessment (day 28). Top: Clone B directly derives from clone A, and therefore possesses all somatic mutations found in clone A, plus additional mutations gained in clone B; Middle: Clone B is completely independent of clone A and thus has a completely different set of somatic mutations; Bottom: Clone B and clone A share a common precursor, hence clone A and B share a common set of somatic mutations found in their common precursor, but each clone has also gained additional mutations of its own. **B** Leukaemia phylogeny in patient P058, delineating the diagnostic clone A (blue open circle) and the refractory clone B (red open circle). Both clones share a precursor clone (black filled circle). Importantly, clone B was present at diagnosis as a minor clone (small red filled circle). **C** Phased B allele frequency (BAF) of heterozygous single nucleotide polymorphisms (SNPs) on chromosome 9 (left column) and chromosome 17 (right column) in whole genome sequencing (WGS) of P058 samples at day 0 (top row) and day 28 (bottom row). Each dot denotes a SNP: major allele (orange), minor allele (black) and SNPs lying outside of the copy number altered segment (grey). **D** Phased BAF of SNPs on

chromosome 9 (left column) and chromosome 17 (right column) in single cell transcriptomes of P058 blasts at day 0 (top row) and day 28 (bottom row). Each dot denotes a SNP where reads have been aggregated across all blasts in that sample: major allele (orange), minor allele (black) and SNPs lying outside of the copy number altered segment (grey). **E** UMAP (uniform manifold approximation and project) showing the expression of *ZBTB16* in day 0 and day 28 blasts from P058. *ZBTB16* expression is largely absent in the day 0 blasts, apart from a small cluster indicated by an arrowhead, whereas the day 28 blasts are strongly expressing *ZBTB16*. **F** UMAPs showing T-cell receptor (TCR) gene usage in day 0 and day 28 blasts from P058, determined by TRUST4 analysis of single cell mRNA sequencing data: TCR genes used by clone A (blue) and clone B (red). **G** UMAPs showing the presence of specific copy number changes (posterior probability ≥0.95) in day 0 and day 28 blasts from P058: chromosome 9 of the precursor clone (black) and chromosome 17 of clone A (blue). **H** UMAPs of day 0 and day 28 blasts from P058, showing the presence of at least one single nucleotide variant (SNV) associated with the precursor clone (black), clone A (blue) and clone B (red). Source data are provided as a Source Data file.

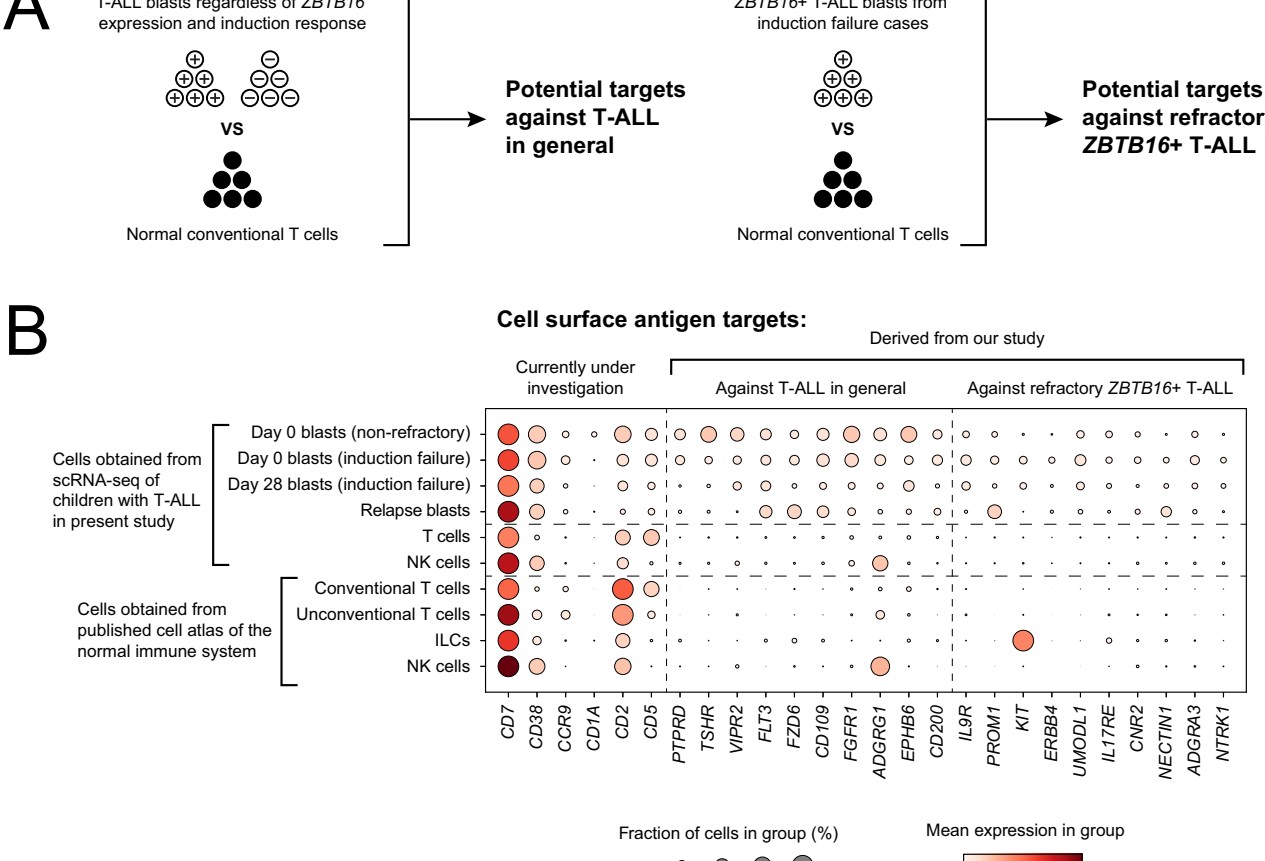

**Fig. 6 | Potential cell surface targets in T-ALL. A** Derivation of two lists of potential antigen targets for (i) T-ALL blasts in general and (ii) refractory *ZBTB16*+ blasts specifically. Differential expression analysis between blasts and normal cells across our discovery and validation scRNA-seq cohorts provided blast-specific genes that were further filtered for cell surface targets. **B** Dotplot showing the expression of potential cell surface targets on cells from our T-ALL cohort (day 0 blasts, day 28 blasts, relapse blasts, and normal T cells and NK cells), as well as cells

from a published single-cell atlas of the normal immune system[45] which include: conventional T cells (T CD4/CD8, T naive/CM CD8, T naive/CM CD4, T naive/CM CD4 activated, Tfh, Tregs, T effector/EM CD4, Trm Th1/Th17, Trm gut CD8, Tem/emra CD8, Trm/em CD8), unconventional T cells (MAIT, Trm Tgd, Tgd CRTAM+), ILCs, and NK cells. Cell surface antigen targets include those derived from our study against T-ALL blasts in general and against refractory *ZBTB16*+ T-ALL blasts, as well as those which are currently under investigation[40–44].

did not confine our study to a particular T-ALL (e.g. ETP T-ALL) or enriched (and thus biased) samples for specific blast populations. It may be that other refractory leukaemias, including B-ALL and myeloid leukaemia, may also be underpinned in part by aberrant blasts which have eluded discovery thus far.

Overall, our findings raise the possibility of predicting poor outcomes at diagnosis. If prospective studies, which will have to utilise single cell assays (e.g. flow cytometry), confirm that non-canonical blasts predict high-risk disease, we would have a powerful diagnostic tool at our disposal to enable an early risk-adapted approach to the

treatment of childhood T-ALL. We would therefore suggest that our findings warrant urgent investigation in prospective clinical cohorts.

## Methods

### Sample acquisition and ethics statement

We accessed tissues from studies approved by UK NHS research ethics committees. The tissue sources were: VIVO Biobank (National Research Ethics Service reference 16SW0219; VIVO project number 23-VIVO-17); Great Ormond Street Haematology Cell Bank (National Research Ethics Service reference 16/LO/0960); samples from the diagnostic archives of Great Ormond Street Hospital (National Research Ethics Service reference 16/EE/0394). Patients or guardians provided informed written consent for participation in this study as stipulated by the study protocols. To minimise the risk of patient identification, age was reported as broad categories. Participants were not compensated.

### Single cell RNA-seq and single cell TCR-seq

Cells were retrieved from fresh cell suspensions or from viably frozen cell suspensions and enriched for viable cells using the Dead Cell Removal Kit (Miltenyi) as per manufacturer's instructions. Single cell droplet suspensions were prepared using the 10x Chromium Controller (10x Genomics), followed by reverse transcription and cDNA amplification using either the Single Cell 5' PE kit (10x Genomics) for the discovery cohort or the Single Cell 5' R2-only kit (10x Genomics) for the validation cohort to generate single cell RNA sequencing (scRNA-seq) libraries. TCR sequences for single cell TCR sequencing (scTCR-seq) were amplified using the Chromium Single Cell Human TCR Amplification kit (10x Genomics). Libraries were sequenced on the NovaSeq 6000 platform.

### Alignment, quantification and quality control of scRNA-seq

Raw reads were processed using Cell Ranger (v7.0.0)[47], where reads were aligned to the GRCh38 human reference genome to generate a table of raw counts per cell. Ambient mRNA contamination was removed with SoupX (v1.6.2)[48] and doublets were removed with Scrublet (v0.2.3)[49]. Single cells were filtered to retain only cells containing >300 expressed genes, >1000 total read count and <15% reads mapping to mitochondrial genes.

### Normalisation, clustering and annotation of scRNA-seq

Count matrices were processed using Scanpy (v1.10.0)[22], which included normalisation to 10,000 total counts and log-transformation. This was followed by highly variable gene selection, principal component analysis (PCA), uniform manifold approximation and projection (UMAP) and clustering using the Leiden algorithm. Batch correction was not performed to ensure that biological variation between individual leukaemias was preserved. Cell clusters were annotated based on their expression of canonical marker genes of haematopoietic cell types and T-ALL immunophenotype markers used in clinical diagnostic panels (Supplementary Figs. 1, 4 and Supplementary Data 1, 3).

### Differential expression analyses

In our initial analysis, we looked for genes that were upregulated in day 28 blasts from patients P058 and P030 (Fig. 1D). This analysis was performed with Scanpy's *rank_genes_groups* function using a Wilcoxon rank-sum test, which identified genes upregulated in either cluster compared to all other blasts (Supplementary Data 4, 5).

To leverage the robustness of bulk RNA differential expression analyses, we compared groups of clusters using a pseudo-bulk approach. Pseudo-bulk differential expression analyses were used in the derivation of the *ZBTB16*-positive blast gene module.

To obtain pseudo-bulk samples, counts were aggregated across each cell population within each sample and were kept if they had a library size >100,000. Genes were retained if they had a logCPM of at least 0.5 in least 10% of samples. EdgeR (v3.42.4)[50] was used to log-

normalise the pseudo-bulk samples and perform differential expression analysis. A quasi-likelihood framework was applied to identify differentially expressed genes in each analysis, using $p$ value threshold of 0.05 and Benjamini–Hochberg $p$ value correction.

We also performed proportional $t$-tests on the proportion of *ZBTB16*+ blasts present in induction failure and responsive patients. For this, we used the eBayes statistical framework proposed in the *propeller* R package[51] that tests logit-transformed proportions to identify if a cell type is enriched in a particular condition. We performed this difference in proportions test for day 0 refractory blasts ($p = 2.6 \times 10^{-7}$), refractory blasts from all other timepoints ($p = 4.4 \times 10^{-9}$), and for day 0 blasts from patients who went on to have an MRD of 1–5% ($p = 0.01$). Each comparison was calculated relative to day 0 responsive patients and was performed on the original and extension cohorts combined.

### Derivation of gene module for bulk RNA-seq scoring

To derive the *ZBTB16*-positive blast gene module, two pseudo-bulk analyses were performed with edgeR:

(1) We aggregated blasts per sample and used spline fitting to identify genes that linearly increase across the three sample groups (day 0 responsive, day 0 induction failure, and day 28 induction failure). Significant genes from this analysis will be lowly- or non-expressed in day 0 responsive cells, have a higher expression in day 0 non-responsive cells, and increase again at day 28, representing genes that are associated with induction failure blasts and increase across the timepoints.

(2) We aggregated *ZBTB16*+ and *ZBTB16*− cell populations separately, per sample, to identify genes that are upregulated in *ZBTB16*+ induction failure blasts compared to *ZBTB16*− responsive blasts. Significant genes from this analysis represent genes that are upregulated in *ZBTB16*+ blasts from non-responsive patients compared to *ZBTB16*− blasts from responsive patients.

Both analyses were performed as described in the methods section above (Differential expression analysis). The intersection of significant genes from these analyses, subject to further filtering, formed the gene module (Supplementary Data 6). Additional filtering was based on:

(1) The proportion of day 0 responsive cells each gene is expressed in to remove genes that are not specific to *ZBTB16*+ cells. Genes were retained if they were expressed in <3% of cells in at least six responsive patients.

(2) The proportion of *ZBTB16*+ induction failure cells each gene is expressed in to retain genes that are specific to *ZBTB16*+ cells. Genes were retained if they were expressed in >2% of cells in at least four induction failure patients.

(3) The effect size of Scanpy module score with the exclusion of each gene, in order to only keep highly-expressed genes that maximally impact module scoring.

### Identification of cell surface targets

We derived two gene lists that represent potential targets for either *ZBTB16*+ cells or T-ALL cells more broadly. To do this, we performed pseudo-bulk differential expression analyses to obtain genes upregulated in blasts relative to each patient's own normal T cells. For the *ZBTB16*+ cell-specific list we used non-responsive *ZBTB16*+ blasts only, for the T-ALL list we used all blasts. To ensure these genes were not expressed in normal cell types and were specific to leukaemic cells, we subject both lists to further filtering; we retained differentially expressed genes that were expressed in:

(1) <15% of day 0 responsive cells (*ZBTB16*+ list only).

(2) > 10% of *ZBTB16*+ cells (*ZBTB16*+ list only).

(3) > 15% of blasts (T-ALL list only).

(4) <10% of cells in at least 10 conventional T cell types from a single-cell atlas of the normal immune system[45] (both lists).

We then subset to genes that are annotated as cell surface proteins in Gene Ontology (under GO:0009897)[52,53].

## Alignment, quantification, and analysis of bulk RNA-seq

Reads were aligned to the GRCh38 reference human genome using STAR (v2.7.10)[54] and quantified with featureCounts (v2.0.2)[55] to provide a raw count table. Genes without a corresponding HGNC ID were removed. Processing of raw counts was performed using edgeR (v3.42.4)[50], which includes gene filtering using the *filterByExpr* function, library-size normalisation and log-transformation to provide log count per million (logCPM) values. Gene module scoring of all bulk RNA samples was performed using singscore (v1.20)[56].

## Bulk RNA-seq analysis of validation cohorts

Two independent bulk RNA-seq cohorts were used in this study: (1) a cohort of 52 patients from the Princess Maxima Center (PMC) and (2) a cohort of 1335 patients who were treated on a clinical trial of the Children's Oncology Group (COG AALL0434)[17].

For the PMC cohort, raw counts were processed into logCPM with edgeR (v3.42.4)[50] as described above except that genes were retained only if they achieved an average logCPM of 0.5 across at least 10% of samples.

For the COG AALL0434 cohort, raw counts for 1335 individuals were downloaded from the Synapse portal (https://www.synapse.org/Synapse:syn54032669/wiki/627818; accessed September 2024). Processing of raw counts into logCPM was performed using edgeR (v3.42.4)[50]. We excluded samples with fewer than 60% blasts (giving 1175 samples) and retained genes with an average logCPM of at least 0.5 in at least 10% of samples.

Survival analyses were performed with the survival R package (v3.5)[47,57] using overall survival and event-free survival as defined by the authors[17]. Patients were grouped into the top and bottom 33.33% according to either *ZBTB16* expression or module score, and these were provided as categorical variables to the Kaplan-Meier model. For Cox proportional hazards modelling, we used both *ZBTB16* and ETP as variables in each model; *ZBTB16* was included either as a continuous variable (logCPM) or the tertiles described above and the ETP variable either included or excluded near-ETP patients. For comparison to BMP-like signatures, we chose to include *ZBTB16* and BMP-like module scores as scaled continuous variables to enable direct comparison between both phenotypes across all patients.

## Flow cytometry to ascertain ETP status

Samples were analysed by flow cytometry to ascertain ETP status as per the reported definition. Once thawed, cells were washed and suspended in flow cytometry buffer (PBS with 2% fetal bovine serum) and stained with antibodies at 4 °C in the dark for 30 min. Antibodies used were CD1a BV421, CD2 FITC, CD3 BV785, CD4 FITC, CD5 PerCP-Cy5.5, CD7 APC, CD8 BV510, CD11b PECy7, CD13 PE, CD15 BV510, CD33 BV605, CD34 PerCP-Cy5.5, CD45 AF700, CD117 BV785 and HLA-DR BV421 (Supplementary Data 16). Samples were run on the BD Fortessa X-20 flow cytometer using BD FACSDiva version 9 (BD Biosciences) and analysed using FlowJo version 10 (BD Biosciences).

## Intra-cytoplasmic flow cytometry assessment of ZBTB16

Once thawed, cells were fixed and permeabilised using the BD Cytofix/Cytoperm™ Fixation/Permeabilization Kit as per manufacturer's instructions. Briefly, cells were washed and suspended in flow buffer and stained with antibodies to surface antigens, including CD3 BV785 (Biolegend), CD45 AF700 (Biolegend) and CD7 APC (Thermo Fisher Scientific) for 30 min at 4 °C. Cells were washed and resuspended in Fixation/Permeabilization solution for 20 min at 4 °C. Cells were then washed and resuspended in BD Perm/Wash™ buffer containing the ZBTB16 (PLZF) PE antibody (BD) for 30 min at 4 °C. Control cells were incubated with an appropriate Mouse PE isotype control (BD).

Following a final wash, cells were resuspended in flow cytometry buffer and run on a BD Fortessa X-20 flow cytometer. Results were analysed using FlowJo version 10 (BD Biosciences) with ZBTB16 positivity gated using the PE isotype control.

## Normal-to-leukaemia transcriptome comparison

A previously described logistic regression analysis[28] was used to test the probability that normal lymphocyte cell states, derived from the human fetal immune atlas[21], resemble our *ZBTB16*+ and *ZBTB16*− blasts. A logistic regression model was trained on single-cell transcriptomes from both blast populations using the *cv.glmnet* function in R, with elastic mixing parameter alpha set to 0.99 for strong regularisation. Neural crest cells from the human fetal adrenal gland[58] were included in the training data as control. This model was used to obtain probabilistic similarity scores for conventional T cells, unconventional T cells, natural killer (NK) cells and innate lymphoid cells (ILC). A similar logistic regression method was used to compare blasts from the validation scRNA-seq cohort to *ZBTB16*+ and *ZBTB16*− blasts from the discovery scRNA-seq cohort.

## Whole genome sequencing and variant calling

Short-insert (500-bp) genomic libraries were constructed and 150-bp paired-end sequencing clusters were generated on the NovaSeq 6000 platform using standard PCR library generation protocol. DNA sequencing reads were aligned to the GRCh38 Ensembl 103 reference genome using the Burrows-Wheeler Alignment tool (v0.7.17)[59].

All classes of somatic variants were called using the extensively validated pipeline of the Wellcome Sanger Institute, built on the following algorithms: CaVEMan (v1.18.2)[60] for base substitutions, Pindel (v3.10.0)[61] for insertions/deletions, ASCAT (v4.5.0)[62] and Battenberg (v3.5.3)[63] for copy number alterations, and BRASS (v6.3.4)[64] and GRIDSS2 (v2.13.1)[65] for structural variants. Copy number alterations in samples that did not have a matched germline sample were called using PURPLE (v3.8.4)[66].

## Alignment and annotation of T-cell receptor sequences

The T-cell receptor (TCR) gene usage of blasts was called using two different methods (Supplementary Data 7). In the first method, raw reads from single-cell TCR sequencing (scTCR-seq) were aligned to the Cell Ranger VDJ reference (v7.0.0), using Cell Ranger (v7.0.0)[47], generating a FASTA file of TCR sequences. The TCR sequences were then annotated by Dandelion (v0.3.6)[67] to give VDJ gene usage calls.

Since scTCR-seq only captures the TCR αβ regions (*TRA/TRB*), an orthogonal method, TRUST4 (v1.1.0)[68], was used to recover the TCR γδ regions (*TRG/TRD*), as well as to validate the *TRA/TRB* calls from Dandelion. TRUST4 extracts candidate reads from the scRNA-seq BAM file, assembles them into TCR sequences and annotates VDJ gene calls.

## Detecting copy number alterations in scRNA-seq

The detection of copy number alterations in single lymphoblast transcriptomes from patient P058 scRNA-seq was performed using alleleIntegrator (v0.9.1)[39]. In brief, heterozygous single-nucleotide polymorphisms (SNPs) were identified from WGS of its remission sample. Phasing of heterozygous SNPs across copy number segments with allelic imbalance was performed, using WGS of the high tumour purity sample at day 0. The number of reads supporting the major/minor allele across each copy number segment was calculated in each single-cell transcriptome. Finally, the posterior probability of each cell harbouring the altered or normal copy number state for each copy number segment was calculated.

## Reconstructing the leukaemia phylogeny of patients P058 and P030

The leukaemia phylogeny of patient P058 across day 0 and day 28 was reconstructed using multiple lines of evidence including driver

mutations, copy number alterations, TCR gene usage and base substitutions. The detection of driver mutations, copy number alterations and TCR gene usage have been described earlier.

Following the standard post-processing filters of CaVEMan, base substitutions had to meet the following criteria: (1) median alignment score of reads supporting a mutation (ASMD) ≥ 140; (2) fewer than half of the reads supporting the variant were clipped (CLPM = 0); (4) not found in loci of low depth or high depth (3) not within 10-bp of any insertion/deletion called by Pindel (including non-PASS ones). Furthermore, site-specific error rates were calculated by interrogating the same sites in 32 unmatched normal blood samples from a previous study[69], and substitutions that were indistinguishable from background noise were rejected[70]. All substitutions post-filtering were visually inspected on the IGV genome browser[71]. The final catalogue of base substitutions is found in Supplementary Data 14.

The variant allele frequencies (VAF) were re-calculated across the day 0 and day 28 samples, with a minimum cutoff for read mapping quality (30) and base quality (25). The cancer cell fractions (CCF) in the two timepoints were then calculated by considering the VAF, the copy number profile and estimated blast percentage[72].

### Statistics and reproducibility
No statistical method was used to predetermine sample size. Bulk RNA sequencing samples from the COG AALL0434 cohort, which had less than 60% blasts, were excluded from analysis. Experiments were not randomised. Investigators were not blinded to any features of the dataset. Sex and/or gender were not considered in the study design. Analyses were performed using either R (v4.3.3) or Python (v3.11.8).

### Reporting summary
Further information on research design is available in the Nature Portfolio Reporting Summary linked to this article.

## Data availability
Raw sequencing data (DNA, bulk mRNA and single cell mRNA) are available through the European Genome-Phenome Archive (EGA) under the following accession codes: EGAD00001009058 (DNA and bulk mRNA sequencing of discovery cohort), EGAD50000001128 (bulk mRNA and single cell mRNA sequencing of discovery cohort), EGAD00001015727 (DNA sequencing of validation cohort, as well as day 28 samples for P058 and P030), and EGAD00001015673 (single cell mRNA sequencing of validation cohort). Data on EGA are accessible on application to the EGA website (https://ega-archive.org/). Processed and raw gene expression counts for single-cell mRNA sequencing of the discovery and validation cohorts are freely available on the CellxGene repository (https://cellxgene.cziscience.com/collections/962df42d-9675-4d05-bc75-597ec7bf4afb). Raw sequencing data and gene expression counts of bulk mRNA sequencing of the Princess Maxima Center cohort are available from Frank N. van Leeuwen (f.n.vanleeuwen@prinsesmaximacentrum.nl). Gene expression counts of bulk mRNA sequencing of the COG AALL0434 cohort (Pölönen et al.[17]) are available for download from the Synapse portal (https://www.synapse.org/Synapse:syn54032669/wiki/627818). Seurat counts for single cell mRNA sequencing data from Xu et al.[34] were provided by the authors and raw data are available for download through dbGaP under the accession number phs003432 as part of the Childhood Cancer Data Initiative. Source data are provided with this paper.

## Code availability
Code used in this study is available here (https://github.com/BehjatiLab/TALL_refractory).

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

## Acknowledgements

Some samples and data used in this study were provided by the Children's Cancer and Leukaemia Group (CCLG) Tissue Bank (now part of VIVO Biobank), supported by Cancer Research UK and Blood Cancer UK

(Grant no. CRCPSC-Dec21\100003). We are indebted to the children and their families who participated in this research. This study was funded by the Wellcome Trust (institutional grant, 108413/A/15/D, and personal fellowship to S.B.) and Cancer Research UK (Clinician Scientist Fellowship funding to D.O'C.). Further funding from Cancer Research UK and Children with Cancer via a Children and Young People's Innovation Award facilitated additional sample collection and analysis through the REVEALL project. Work done at the Cancer Research UK City of London Centre Single Cell Genomics Facility and University College London Cancer Institute Bioinformatics Hub was supported by the Cancer Research UK City of London Centre Award (CTRQQR-2021/100004). We received additional funding from the Harley Staples Cancer Trust, the Agency for Science, Technology and Research (personal fellowship to B.S.J.L.), the Wenner-Gren Foundations (personal fellowship to A.W.), the Pessoa de Araujo family (personal fellowship to A.H.), The Little Princess Trust (personal fellowship to T.D.T.), the EMBO long-term fellowship (ALTF 172-2022, personal fellowship to T.H.H.C.), the National Institutes of Health Gabriella Miller Kids First Pediatric Research (X01HD100702, D.T.T.) and the Great Ormond Street Children's Charity (professorship to M.R.M.). This research was supported by the NIHR GOSH Biomedical Research Centre and NIHR Cambridge Biomedical Research Centre (NIHR203312). The views expressed are those of the authors and not necessarily those of the NHS, NIHR or Department of Health.

## Author contributions

Conceptualisation: S.B. and D.O'C. Experiments: G.Bl., R.T., C.P., T.O., O.W., S.A. and D.O'C. Data generation: G.Bl., A.H., C.P., T.O., P.P., C.G.M., D.T.T., J.X., K.T., M.H., L.K., F.N.V.L., G.Be., M.R.M., J.B., S.B. and D.O'C. Data analysis: B.S.J.L., H.J.W., M.K.T., N.D.A., A.W., T.D.T., H.L.-S., T.H.H.C. and L.J. Supervision: S.B. and D.O'C. Manuscript writing: B.S.J.L., H.J.W., M.K.T., A.H., L.J., S.B. and D.O'C.

## Competing interests

D.T.T. receives research funding from BEAM Therapeutics and Neoimmune Tech. D.T.T. serves on advisory boards (unpaid) for Amgen, BEAM Therapeutics, Novartis, Jazz, J&J Innovation, Pfizer, Sobi, Servier and Syndax. D.T.T. holds patents or has patents pending: "Biomarkers predictive of cytokine release syndrome" (US11747346) and "Compositions and methods comprising anti-CD38 chimeric antigen receptors" (US20250121004, pending). C.G.M. receives research funding from AbbVie and Pfizer. C.G.M. has a consulting role with Amgen and Illumina. C.G.M. receives royalties from Cyrus. The remaining authors declare no competing interests.
