## [Transparent Peer Review file · Nature Communications]

A non-canonical lymphoblast in refractory childhood T-cell leukaemia

Corresponding Author: Professor Sam Behjati

Version 0:

Reviewer comments:

Reviewer #1

(Remarks to the Author)

The authors addressed my comments appropriately. Thank you very much and congratulations for this nice work

Reviewer #3

(Remarks to the Author)

The authors have addressed the key concerns by expanding their analysis to include an additional 37 children, comprising 55 samples, which were analysed using single-cell mRNA sequencing and multiparametric flow cytometry. This extended cohort supports their conclusion that ZBTB16 is a novel marker of these 'non-canonical' refractory cells and has further allowed the potential identification of downstream targets for further investigation. They also provide further statistical analysis that highlight the independent prognostic value of ZBTB16 vs. current clinical parameters/subtype classification.

While the authors do not provide functional data to determine whether ZBTB16 plays a direct role in treatment resistance — or is merely a marker— this point, raised by all three reviewers, is acknowledged and discussed in detail in the manuscript discussion. While this would clearly strengthen the paper, I accept the authors' response that such functional studies would constitute a separate line of investigation and do not detract from the current study's significance.

Overall, I believe the authors have adequately addressed the reviewers' concerns. The study presents important and novel findings that may lead to new strategies for stratifying and potentially targeting residual refractory T-ALL blasts—a patient subgroup for whom effective therapies are currently limited. This represents a meaningful advancement in the field and warrants publication.

Reviewer #4

(Remarks to the Author)

The authors have addressed the main comments in their revision.

Importantly, the functional role of ZBTB16 in treatment resistance remains unclear and merits further investigation to advance potential translational applications.

A non-canonical lymphoblast in refractory childhood T-cell leukaemia

Response to Reviewers

Reviewer	Comment	Response
Reviewer #1	The authors addressed my comments appropriately. Thank you very much and congratulations for this nice work	We thank the Reviewer for their comments.
Reviewer #3	The authors have addressed the key concerns by expanding their analysis to include an additional 37 children, comprising 55 samples, which were analysed using single-cell mRNA sequencing and multiparametric flow cytometry. This extended cohort supports their conclusion that ZBTB16 is a novel marker of these 'non-canonical' refractory cells and has further allowed the potential identification of downstream targets for further investigation. They also provide further statistical analysis that highlight the independent prognostic value of ZBTB16 vs. current clinical parameters/subtype classification. While the authors do not provide functional data to determine whether ZBTB16 plays a direct role in treatment resistance—or is merely a marker—this point, raised by all three reviewers, is acknowledged and discussed in detail in the manuscript discussion. While this would clearly strengthen the paper, I accept the authors' response that such functional studies would constitute a separate line of investigation and do not detract from the current study's significance. Overall, I believe the authors have adequately addressed the reviewers' concerns. The study presents important and novel findings that may lead to new strategies for stratifying and potentially targeting residual refractory T-ALL blasts—a patient subgroup for whom effective therapies are currently limited. This represents a meaningful advancement in the field and warrants publication.	We thank the Reviewer for their comments.
Reviewer #4	The authors have addressed the main comments in their revision. Importantly, the functional role of ZBTB16 in treatment resistance remains unclear and merits further investigation to advance potential translational applications.	We thank the Reviewer for their comments.